# Cognition-Supervised Saliency Detection: Contrasting EEG Signals and Visual Stimuli

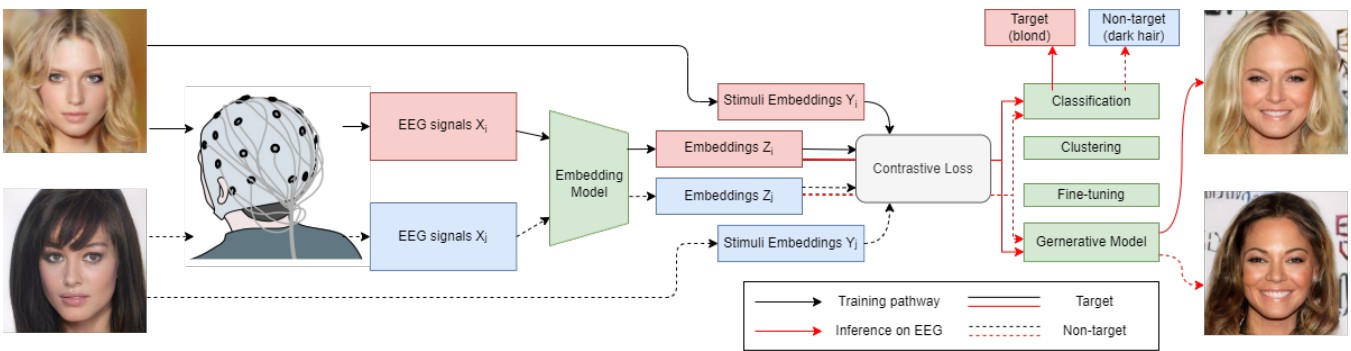

**Figure 1: Illustration of cognition-supervised visual saliency detection and its applications in downstream tasks. EEG responses to visual stimuli are used to train embedding models through CLIP loss by contrasting EEG with stimuli image representations. The learned EEG embeddings are applied in downstream tasks such as clustering, classification, fine-tuning personalized models, and conditioning generative models. No manual annotation is needed during training or inference.**

## ABSTRACT

Understanding human assessment of semantically salient parts of multimedia content is crucial for developing human-centric applications, such as annotation tools, search and recommender systems, and systems able to generate new media matching human interests. However, the challenge of acquiring suitable supervision signals to detect semantic saliency without extensive manual annotation remains significant. Here, we explore a novel method that utilizes signals measured directly from human cognition via electroencephalogram (EEG) in response to natural visual perception. These signals are used for supervising representation learning to capture semantic saliency. Through a contrastive learning framework, our method aligns EEG data with visual stimuli, capturing human cognitive responses without the need for any manual annotation. Our approach demonstrates that the learned representations closely align with human-centric notions of visual saliency and achieve competitive performance in several downstream tasks, such as image classification and generation. As a contribution, we introduce an open EEG/image dataset from 30 participants, to facilitate further research in utilizing cognitive signals for multimodal data analysis, studying perception, and developing models for cross-modal representation learning.

## CCS CONCEPTS

• **Human-centered computing** → **HCI design and evaluation methods**; • **Computing methodologies** → **Cognitive science**.

## KEYWORDS

Electroencephalography, EEG, Contrastive Learning, Generative Modeling, Neuroimaging

## 1 INTRODUCTION

Human cognition excels at detecting salient information from various media, rapidly identifying what is important for an individual in a specific context or multimedia experience. This innate ability to discern salient parts of stimuli is crucial for a range of multimedia applications, including generating personalized content recommendations, interfaces, and multimedia experiences. However, replicating this capability in machines, particularly in a way that reflects an individual's perception of semantic saliency, remains a significant challenge. Traditional machine learning approaches often rely on large datasets of implicitly obtained signals, such as click data [27, 37, 50] or dwell time [59], from platforms that expose information to their users, such as social media or video streaming services. These data, often combined with visual information under a supervised learning framework [58], serve as proxies for cognitive responses to salient features. However, these supervision data may not always capture the nuanced cognitive preferences of individuals and is dependent on availability of behavioral data, such as clicks.

In this work, we propose an alternative approach, *cognition-supervised saliency detection*, to capture human cognitive responses to visual information without reliance on manual labels or behavioral data. Our method utilizes natural human cognitive reactions evoked by visual perception of media content and measured via

electroencephalogram (EEG). Specifically, participants are exposed to visual information while their EEG signals evoked in response to perception are recorded. These EEG data are then employed in a self-supervision framework to learn representations of salient semantic visual features eliciting variations in cognitive responses.

Integrating brain responses into machine learning has historically been challenging. Prior research on visual saliency detection leveraging brain responses often depended on manually labeled data [12, 44, 62] or the fine-tuning of models pretrained on such data [14, 18, 48, 53]. Unsupervised methods, conversely, tend to underperform with brain data due to its inherent noise and complexity [40]. In scenarios such as supervised binary classification using only single-trial data with complex visual stimuli, accuracy typically remains modest – for instance, 0.78 for differentiating human faces from objects [33], about 0.7 for within-subject and roughly 0.4 for cross-subject comparisons [33, 63], 0.708 for ImageNet subsets [2], and under 0.6 for text stimuli [19]. Moreover, certain earlier studies addressing similar challenges have been criticized for relying on confounded datasets [34]. As a result, using brain signals as a direct source of supervision for machine learning models is facing two fundametal problems: achieving a performance that has utility for practical applications and learning without manually labeled data.

Our approach tackled both of these problems. It learns representations of semantic visual saliency as perceived by the brain, utilizing unlabeled EEG data contrasted with visual stimuli for supervision. The model is crafted to differentiate between target and non-target saliency, based on participants' brain responses.

Utilizing this model, we explore two primary research questions:

**RQ1:** Can representations of semantic saliency be directly learned from EEG data as a supervision signal?

**RQ2:** Do the learned representations accurately capture the salient features in downstream tasks?

With our experiment results, we demonstrated that the learned representations reflect the desired semantic visual saliency better than any single modality representations, and linear classifiers built on top of them have performance comparable to supervised models. We also tested our method in personalization scenarios where a base model can flexibly adapt to a small amount of personal data. In a generative downstream task, the learned representations are used to successfully condition the generated images to fully match the semantic saliency given in the task.

Additionally, to further research in this area, we are releasing an open, anonymized EEG dataset from 30 participants. This dataset, designed with specific semantic saliency detection tasks for multimedia content, aims to promote advancements in cognition-supervised models.

In summary, our main contributions are:

- Introducing a novel approach for contrastively training models with cognitive EEG responses to visual multimedia stimuli, to learn representations of semantic visual saliency.
- Releasing a new open and anonymized EEG dataset from 30 participants, complete with a comprehensive codebase, to encourage research in cognition-supervised models for multimedia applications.

## 2 RELATED WORK

In recent years, the integration of brain signals with machine learning has garnered considerable attention for its potential to enhance both the performance and interpretability of models. Among the array of brain-computer interface devices, electroencephalography (EEG) signals stand out as a favored modality, offering rich, albeit noisy, data for supervised machine learning models. EEG is valued for its non-intrusive nature, high temporal resolution, and cost-effectiveness.

However, EEG signals are inherently challenged by limited spatial resolution and susceptibility to artifacts and noise from subject movements, which can significantly impede the efficacy of EEG-based machine learning models. This is particularly prominent for models addressing cognitive processes like visual semantic saliency recognition in real-world media content. The modest spatial resolution of EEG complicates the accurate localization of neural activity tied to visual cognition, and noise can further obscure the cognitive signals of interest.

Decoding EEG signals has enabled a wide range of applications, including emotion recognition [3, 24] for affective multimedia experiences, mental workload assessment [4, 46] for adaptive user interfaces, and multimedia content understanding [29, 42]. These applications are grounded in supervised EEG classification models, which facilitate the effective use of brain data across various contexts.

However, traditional supervised machine learning approaches rely on manual annotations, presenting challenges related to cost of training the models and subjectivity of the training data. Manual annotations, requiring domain experts to label vast quantities of multimedia data, are both time-consuming and resource-intensive. Moreover, the subjectivity inherent in human annotations can lead to inter-annotator variability, undermining the reliability and consistency of annotations, especially for subjective phenomena like emotions and cognitive responses to multimedia content. This may force "one-size-fits-all" models and ignore the need for personalized models of human cognition.

To address these limitations, there is a growing need for unsupervised and self-supervised approaches that leverage EEG as supervisory signals to train machine learning models. Recent research has explored the direct use of brain signals as supervisory signals for machine learning models [8, 9, 16, 32]. Self-supervised learning with EEG data may provide potentially more objective and quantifiable measures of brain activity, leading to more reliable and cost-effective annotations compared to traditional methods requiring expert knowledge for manual annotation. Moreover, the real-time capture of brain responses enhances the adaptability and robustness of machine learning models, allowing them to dynamically respond to changes in brain states during perception of digital information.

A series of earlier studies on EEG-based image reconstruction suffer from confounded EEG data due to specific experimental block designs. This includes the EEG-GAN approach [43, 52], Thoughtviz [54], Brain2image [29], EEG-ChannelNet [42], and numerous subsequent research on the same datasets such as EEG2IMAGE [51], DM-RE2I [61], NeuroGAN [38], and GDN-GAN [30]. To this end, subsequent analyses [1, 2, 34] have identified a critical flaw in these

approaches: the block design in data collection introduces temporal correlations between the presentation order of stimulus class and the experiment duration. Attempts to replicate these studies have suggested that models were learning to recognize the order of stimuli presentation rather than the genuine cognitive reactions to the stimuli [34].

In parallel, contrastive learning methods have gained significant attention in the broader field of machine learning [10, 20–22, 45, 56, 57]. Contrastive learning aims to develop robust and meaningful representations without explicit annotations, by maximizing the similarity between positive pairs (similar samples) and minimizing it between negative pairs (dissimilar samples). The efficacy of contrastive learning has been proven in areas like large language models [45], image embeddings [25], and audio data [47], yet its application in EEG-based machine learning to be paired with multimedia data is still relatively unexplored. Similar contrastive methods have been applied to EEG data for tasks such as sleep stage classification [26], emotion recognition [39], and pathology screening [6]. These methods often rely on carefully designed data augmentation or combinations of transformations. However, identifying effective data augmentation techniques for EEG data in multimedia scenarios remains a challenge. A recent study [26] highlighted how improper transformation choices could significantly reduce test accuracy, from 82.90% to 48.15%.

Moreover, focusing contrastive learning on a single modality may overlook valuable information from other modalities, especially in multimedia applications that typically involve multiple modalities. This challenge was tackled by supervised contrastive learning [20], which enhances image pair augmentation with labeling for better grouping. Another study [57] introduced a hierarchical semantic alignment strategy to assess the semantic similarity between images. Additionally, a multimodal contrastive training method [60] employed multiple loss functions to leverage the intrinsic data structure of each modality.

Our work is inspired by the well-known language supervision approach CLIP [45], which learns representations from paired text and image data to align across two modalities. Similarly, our embedding model aligns representations from paired EEG and visual stimuli data, effortlessly obtained from data collection in multimedia scenarios. Unlike merely decoding EEG signals to categorical or simplistic stimuli, our contrastive approach bridges EEG with high-dimensional multimedia stimuli. In a manner akin to CLIP, which is grounded in natural language supervision, we define our method as cognition-supervised learning.

A recent study [49] investigated non-linear techniques to learn a consistent latent space of joint behavior and neural data across subjects, applying this method to various animal datasets. However, it's crucial to recognize that this data was acquired through intrusive methods, using implanted electrodes or probes, which are more challenging to apply to human subjects in multimedia contexts than non-intrusive EEG signals. This study aimed at movie frame reconstruction, focusing on the sequence of movie frames rather than their content, which restricts its applicability to multimedia applications featuring diverse and dynamic content.

Consequently, a significant research opportunity exists for effective cognition-supervised learning within multimedia applications. To fill this gap, we introduce a novel approach that leverages the contrast between EEG data and multimedia stimuli as a supervisory signal. Our method benefits from label-free learning using EEG data and integrates stimuli information to remain effective even with a limited amount of EEG data, which might be insufficient for self-supervision if relying solely on EEG signals.

## 3 METHODS

### 3.1 Data Collection and Preparation

To explore the feasibility of cognition-supervised learning, we conducted neurophysiological experiments to gather EEG responses to generated visual stimuli. The acquisition of neurophysiological data and subsequent experiments received approval from the ethical review board of social and behavioral sciences at *anonymous organization*, adhering to the Declaration of Helsinki[1]. Informed consent was signed by each participant to acknowledge their rights. Participants were compensated with vouchers for the local cinema.

**Visual Stimuli Preparation.** We selected generated images of faces as visual stimuli, recognizing that humans exhibit strong responses to facial stimuli [55]. Generated images were chosen over real ones to control for variances in semantics and confounding visual features, thereby minimizing brain responses related to recognition effects. This approach ensures a homogeneous dataset that facilitates strict semantic-level evaluation in generative tasks. A random sample of 70,000 images was generated using a progressive GAN[2] [28], pre-trained on the CelebA dataset [35]. The raw images, with a resolution of 1024 by 1024 pixels, were manually screened by researchers to remove images with visual artifacts, such as distorted faces or evident signs of artificiality, to ensure brain responses reflected semantic saliency rather than artifact recognition. These images were categorized into eight groups based on semantic saliency: smiling, not smiling, female, male, young, old, dark hair, and light hair (blond).

**Participants.** Neurophysiological data were collected from thirty participants (self-reported 13 female and 17 male, mean age 28 years (SD = 7.14, Min = 18, Max = 45)) at *anonymous organization*. Participants were healthy with normal or corrected-to-normal vision.

**Apparatus, Tasks, and Procedure** Participants were exposed to eight recognition tasks sequentially, each corresponding to one of the semantic saliency groups (e.g., female, smiling). An elliptic grey frame was used to obscure the backgrounds of all images. EEG data were captured using 32 Ag/AgCl electrodes, placed according to the 10–20 system, and connected to a QuickAmp USB (BrainProducts GmbH, Gilching, Germany) amplifier with a sampling rate of 2,000 Hz. Eye movements were monitored for artifact removal through two pairs of bipolar electrodes positioned near the eyes (1 cm lateral to the left and right canthi, and 2 cm above and below the right pupil).

For each task, stimuli were labeled binary according to their semantic saliency. For instance, in the "smile" task, participants viewed images of smiling (target) and non-smiling (non-target) faces. Participants were asked to mentally note images matching

---

[1]http://www.wma.net/en/20activities/10ethics/10helsinki/
[2]https://github.com/tkarras/progressive_growing_of_gans under the attribution-noncommercial 4.0 international (cc by-nc 4.0) license

the task description without any physical response. Each task iteration presented twenty target and twenty non-target images in a random sequence. Stimuli were displayed using a rapid serial visual presentation (RSVP) method at a 500 ms interval. A demonstration task was conducted before each main task to confirm participants' understanding, where they were to identify images featuring the specified semantic attribute.

**Data Preprocessing.** Post data collection, we applied standard signal cleaning techniques [36] to enhance the signal-to-noise ratio, involving only automatic methods that do not necessitate any additional labeled data. This included applying a band-pass filter within the 0.2–35 Hz frequency range and time-locking epochs from -200 to 900 ms relative to stimulus onset, with baseline correction using the pre-stimulus interval from -200 to 0 ms. Eyeblink artifacts were eliminated using a threshold heuristic, setting the threshold to the 200th largest mean absolute value across epochs and channels, within a [10,80] range. After preprocessing and balancing, an average of 1144 epochs per participant remained. Data aggregated across participants revealed a typical P300 effect, with increased potentials for target stimuli observable from 250 ms post-stimulus onset until 600 ms, as shown in Figure 2. This result validates the anticipated ERP effect on a population average through our experimental setup.

## 3.2 Cognitive Supervision

Cognition-supervised learning is predicated on the fundamental observation that the human brain differentially responds to perceptual stimuli. This principle implies that the contrast between visual stimuli and brain responses can act as a supervision signal, enabling direct learning from preferences reflected in cognitive processes. This contrastive learning framework allows for the development of a loss function that relies solely on EEG data and stimuli, obviating the need for manual annotations.

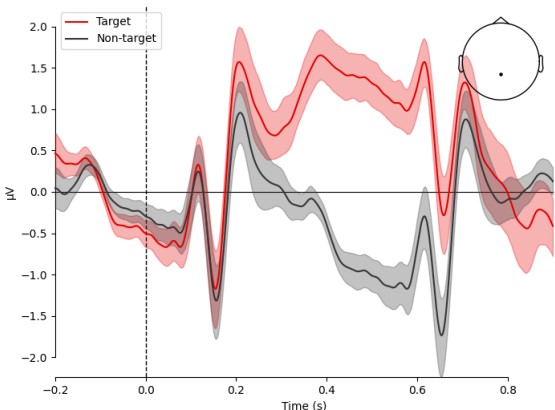

**Figure 2: Average event-related potentials (ERPs) across participants at the Pz electrode for target and non-target stimuli, illustrating a P300 effect.**

To facilitate cognition-supervised learning, we introduce a model that embeds EEG signals. For each stimulus image generated from a latent vector $Y \in \mathbb{R}^L$ (where $L$ is the dimension), we represent the corresponding epoched EEG response as $X \in \mathbb{R}^{C \times T}$, with $C$ denoting the number of channels and $T$ the number of time steps in the epoch.

Initially, we considered employing a regression model $f_{reg} : \mathbb{R}^{C \times T} \to \mathbb{R}^L$ to reconstruct the stimulus vector from EEG inputs. However, this approach tended to overfit to noise and showed limited generalizability. Moreover, reconstructing the entire stimulus vector from EEG is impractical since only salient semantic features and major facial attributes are discernible to participants. Consequently, we adopted a noise contrastive estimation using a CLIP loss [45], aiming to embed semantic saliency as perceived by participants.

Our proposed embedding model $f_{embed} : \mathbb{R}^{C \times T} \to \mathbb{R}^L$ is trained with an EEG signal $X$ and its associated stimulus vector $Y$, alongside a set of negative stimulus vectors $Y_i$ for $i \in \{2, 3, \ldots, N\}$, ensuring $Y_i$ differ from $Y$. The negative set is sampled from the remaining stimuli vectors in the dataset while avoiding duplication of $Y$ and $Y_i$. $Y_1 := Y$ is treated as the positive sample.

The model $f_{embed}$ predicts the probability $\hat{p}_j = \mathbb{P}[Y_j = Y]$ by calculating the dot product between $Z := f_{embed}(X)$ and each $Y_j$, followed by a Softmax function. The probability is defined as:

$$\hat{p}_j = \frac{e^{\langle Z, Y_j \rangle}}{\sum\limits_{j'=1}^{N} e^{\langle Z, Y_{j'} \rangle}} \quad (1)$$

where $\langle \cdot, \cdot \rangle$ indicates the inner product.

$f_{embed}$ is optimized using cross-entropy between the actual probability $p_j$ and the estimated $\hat{p}_j$, where $p_j = 1$ iff $j = 1$, and $p_j = 0$ otherwise. The loss function simplifies to:

$$L_{\text{CLIP}}(p, \hat{p}) = -\langle Z, Y \rangle + \log\left(\sum_{j=2}^{N} e^{\langle Z, Y_j \rangle}\right) \quad (2)$$

## 3.3 Model Structure

To accommodate inter-subject variability while capturing the intrinsic structures of EEG signals, we employ a deep neural network, $f_{embed}$, processing raw, vectorized EEG signals. Additionally, a one-hot encoded vector representing the participant is input to the network, which outputs an embedding vector $Z$ of the same dimension as the stimulus vector $Y$. The network architecture comprises two components: (1) a participant-specific convolution matrix and (2) a series of fully connected layers.

**Fully Connected Layers.** The embedding model features four fully connected layers. The first three layers each have 2048 hidden nodes and utilize a LeakyReLU activation function with $\alpha = 0.3$. Following each of these layers is a Dropout layer with a dropout rate of 0.5. The final layer outputs 512 nodes without any activation function.

**Participant-Specific Matrix.** To accommodate variability across participants within a unified model, we implement a strategy similar to that described in [17]. A participant-specific layer, placed at the outset of the network, contains a trainable $C \times C$ matrix for each participant. This matrix is applied to the vectorized $C \times T$ EEG

signal across channels, initialized near the identity matrix with slight random perturbations.

**Data Augmentation.** To enhance model generalization and mitigate overfitting, we apply random data augmentation during training. Initially, each EEG vector $x \in \mathbb{R}^{C \times T}$ is scaled by a random vector $c \in [0.95, 1.05]^C$. Subsequently, we perform a random crop and resize on the vector to $x' \in \mathbb{R}^{C \times T}$, selecting an interval $[l, r]$ where $l \in [0, \frac{T}{10}]$ and $r \in [T - \frac{T}{10}, T]$. For the validation set, we consistently crop and resize the data within the fixed interval $[\frac{T}{20}, T - \frac{T}{20}]$, excluding the random scaling.

## 4 EXPERIMENTS

In this section, we evaluate the efficacy of cognitive supervision on embedding models through structured experiments. Initially, t-SNE visualization assesses embedding space clustering for target versus non-target saliency. Next, unsupervised clustering examines the embedding space's alignment with target and non-target distinctions. Subsequently, a linear evaluation protocol [5, 13, 15, 23, 31, 41] determines if classifiers on saliency embeddings outperform non-supervised data. We then explore personalized model tuning. Finally, a qualitative assessment is conducted using generative adversarial networks to visualize cognition-supervised predictions.

**Dataset.** Our dataset consists of EEG signal and stimuli vector pairs from 30 participants, comprising a total of 35490 pairs. To ensure that the individual factor is accounted for, we mixed all participant data while retaining a unique participant identifier to apply the participant-specific matrix. For the unsupervised task, we trained and evaluated our embedding model on the entire dataset. For linear classification tasks, we employed 10-fold validation by randomly splitting the dataset into training and testing sets and reported the mean of evaluation metrics. All experiments were repeated three times with different random seeds. To accurately reflect the variability due to methodological randomness rather than distribution differences, results are averaged across the tasks for each run, and standard deviations of these averages are computed and reported alongside all evaluations.

**Hyperparameters and Hardware.** In all experiments, the brain embedding model is trained with Adam optimizers with an initial learning rate $1e - 4$, $\beta_1 = 0.9$, $\beta_2 = 0.999$, and a weight decay $1e - 4$. The mini-batch size is set to 256 in all experiments. We conducted all experiments on Tensorflow with a single Nvidia GeForce RTX 3070 Ti GPU. Each embedding model in the unsupervised clustering experiments, linear evaluation experiments and the base model in the personalized experiments are trained with 500 epochs. The personalized model is fine-tuned by 100 iterations by using the base model, with all other parameters frozen except the participant-specific matrix.

### 4.1 t-SNE Visualization of Embeddings

First, we provide an intuitive understanding of the learned embeddings by visualizing the feature space of EEG, stimuli, and embedding space visualized via t-SNE as shown in Figure 3. The data points are colored according to the target and non-target stimuli. The learned embedding space separates the data significantly better than either of the original modalities. The target and non-target

data points are not linearly separable in either the EEG or stimuli space, but they are clearly separable in the learned embedding space.

### 4.2 Unsupervised Clustering and Classification

The discernible patterns within the t-SNE visualizations inspire us to perform unsupervised clustering to automatically distinguish between target and non-target clusters, leveraging the model's innate capability to differentiate salient features without label assistance.

**Evaluation Procedure.** Employing KMeans with $k = 2$ on the embeddings, we identify two clusters. The one with a higher average P300 effect is designated as the Target cluster ($C_T$), while the other is recognized as the Non-target cluster ($C_N$). This method allows for the autonomous classification of the dataset into meaningful groups that correspond to the stimuli's inherent saliency, without any reliance on explicit labels. We compute the clustering accuracy as the ratio of correctly classified samples to the total number of samples.

**Control Models.** For context, we compare our approach against KMeans clustering applied to (1) stimuli vectors, (2) flattened EEG signals, and (3) their concatenated forms. This comparison highlights the efficacy of our embeddings in capturing cognitive patterns. The highest clustering accuracy over all clusters permutations is reported for control models.

**Results.** Table 1 summarizes the clustering accuracies, underscoring our model's superiority in discerning between target and non-target clusters across various tasks. This outcome confirms the model's effectiveness in capturing participant-perceived salient features purely through unsupervised learning.

### 4.3 Linear Evaluation

**Evaluation procedure.** To evaluate the efficiency of the learned saliency representations, we follow the commonly used linear evaluation protocol, by training a linear classifier on top of the frozen embeddings. The dataset is randomly split into a training set and a testing set with disjoint sets of stimuli. We then train our contrastive embedding model on the training set and then compute the embeddings with frozen model weights. A single-layer binary classifier $C(\cdot) : \mathbb{R}^{512} \rightarrow \{0, 1\}$ is trained on the embeddings from the training set using the explicit labels of stimuli images. The classifier is then evaluated on the test set using the labels with classification accuracy.

**Control models.** To provide a basis for comparison, we also consider three control models as the baseline. The first is a well-known supervised EEGNet [33] structure to estimate the upper limit of performance for the cognition-supervised models and highlights the difficulties of the task. The second is a linear discriminant analysis model (LDA) [7] to estimate the separability of raw EEG signals. Both control models are trained on the raw EEG signals and the explicit labels. The third baseline model is a randomly permuted cognition-supervised EEG classifier to determine a lower bound performance, in which the pairs of EEG signals and stimuli vectors are shuffled so that the pairs are broken.

**Results.** Table 2 shows the mean accuracies of all models for each task. The linear classifiers on saliency embeddings consistently outperform the random baseline and the LDA models, indicating

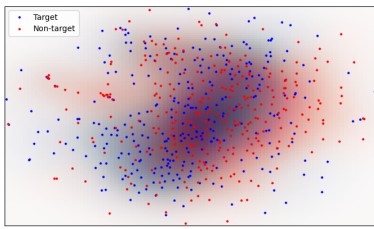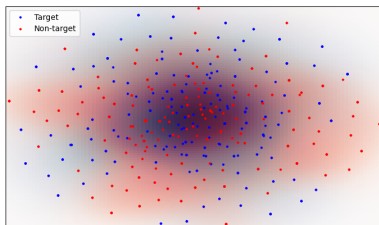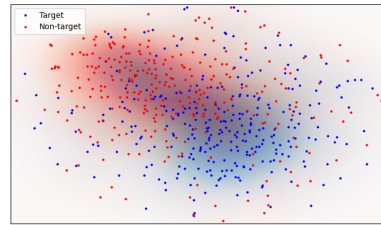

**Figure 3: Visualization of feature space of (left) EEG, (middle) stimuli and (right) embedding space visualized via t-SNE. The learned embedding space separates the data significantly better than either of the original modalities.**

**Table 1: Clustering accuracies on all tasks with different inputs for KMeans.**

| KMeans input | female | male | blond | darkhaired | smiles | nosmile | old | young | Mean |
|---|---|---|---|---|---|---|---|---|---|
| stimuli vectors | 0.57±0.02 | 0.56±0.06 | 0.54±0.01 | 0.53±0.03 | 0.51±0.01 | 0.52±0.01 | 0.52±0.02 | 0.53±0.01 | 0.536±0.011 |
| EEG signals | 0.57±0.01 | 0.51±0.01 | 0.56±0.01 | 0.52±0.01 | 0.57±0.01 | 0.50±0.01 | 0.52±0.01 | 0.55±0.01 | 0.537±0.001 |
| concatenated | 0.55±0.02 | 0.53±0.02 | 0.56±0.03 | 0.56±0.04 | 0.52±0.01 | 0.52±0.01 | 0.53±0.01 | 0.52±0.01 | 0.535±0.006 |
| ours | **0.76**±0.01 | **0.70**±0.01 | **0.71**±0.03 | **0.71**±0.01 | **0.70**±0.02 | **0.67**±0.01 | **0.61**±0.01 | **0.67**±0.01 | **0.691**±0.004 |

**Table 2: Classification accuracies for linear classifiers on top of learned representations and other supervised models. The train/test split is the same for all models.**

| Method | female | male | blond | darkhaired | smiles | nosmile | old | young | Mean |
|---|---|---|---|---|---|---|---|---|---|
| EEGNet | **0.76**±0.01 | 0.67±0.01 | **0.74**±0.02 | **0.71**±0.03 | **0.72**±0.01 | **0.69**±0.01 | **0.64**±0.02 | **0.68**±0.01 | 0.700±0.005 |
| LDA | 0.58±0.01 | 0.54±0.01 | 0.56±0.02 | 0.55±0.02 | 0.56±0.01 | 0.55±0.01 | 0.53±0.01 | 0.55±0.01 | 0.553±0.005 |
| random control | 0.52±0.02 | 0.53±0.02 | 0.53±0.01 | 0.52±0.02 | 0.51±0.01 | 0.51±0.01 | 0.51±0.01 | 0.54±0.01 | 0.519±0.004 |
| ours | **0.76**±0.01 | **0.70**±0.02 | 0.73±0.01 | **0.71**±0.01 | 0.71±0.01 | 0.68±0.01 | 0.63±0.01 | 0.65±0.02 | **0.701**±0.005 |

that the learned embeddings were effective in disentangling semantic features. Furthermore, we observed that the mean accuracy across all tasks is higher than that of the EEGNet, which suggests that the learned embeddings successfully reduced the high dimensionality of raw EEG signals while preserving the saliency perceived by the participant. It is worth noting that, the embedding model is trained without labels and the supervised linear classifier on top of it is expected to have relatively lower performance compared to a completely supervised model, as shown in [11].

### 4.4 Personalized Model Evaluation

**Evaluation procedure.** To extend our model's utility to accurately reflect individual cognitive responses, we evaluated fine-tuned personalized models. For each of the 30 participants, a base model was initially trained using EEG data from the other 29 participants. The target participant's data was then divided into a 5-fold training set and a test set. We froze the base model's weights, with the exception of the participant-specific matrix, which was randomly initialized and then fine-tuned using the single-participant training set. Frozen saliency embeddings from other participants were used to assist in selecting the target cluster.

**Control models.** For comparison, we also evaluated two control models. The first control model is the base model evaluated on the test set without fine-tuning. The target participant matrix is set to

the identity matrix. The second control model was fine-tuned on randomly shuffled training data, breaking the pairs of EEG signals and stimuli vectors.

**Results.** Table 3 presents the mean clustering accuracy on the test set and reports the mean from 5-fold validation across all participants. The personalized models demonstrated a small improved accuracy over the base models on average on all tasks. Our results suggest that with more data available, our method has the potential to further improve. In addition, the base model, which was not trained with personal data, still achieved a high clustering accuracy compared to the random control model. This underscores the zero-shot prediction capability of our embedding model, highlighting its potential to learn robust representations and adapt to the subjective information from individual cognitive signals.

### 4.5 Qualitative evaluation via generative visualization

**Generative Visualization of Salient Features.** In order to provide an intuitive understanding of the inter-subject variations, we visualize the embeddings for a qualitative evaluation. We sample a set $S$ of candidate stimuli vectors which can be the set of stimuli vectors in the training set, or a fresh set of randomly sampled vectors from the noise distribution used in the generative model.

**Table 3: Clustering accuracies of all personalized models.**

| Model | female | male | blond | darkhaired | smiles | nosmile | old | young | Mean |
|---|---|---|---|---|---|---|---|---|---|
| random control | 0.56±0.01 | 0.53±0.01 | 0.55±0.02 | 0.55±0.01 | 0.51±0.01 | 0.52±0.01 | 0.52±0.01 | 0.55±0.01 | 0.537±0.025 |
| base model | **0.74**±0.01 | 0.66±0.01 | **0.70**±0.01 | **0.68**±0.01 | **0.70**±0.01 | 0.65±0.01 | 0.61±0.01 | 0.65±0.01 | 0.673±0.020 |
| personalized model | **0.74**±0.01 | **0.67**±0.01 | **0.70**±0.01 | **0.68**±0.01 | **0.70**±0.01 | **0.66**±0.01 | **0.62**±0.01 | **0.66**±0.01 | **0.682**±0.021 |

**Table 4: Clustering accuracies of model variants in ablation study.**

| Model Variant | female | male | blond | darkhaired | smiles | nosmile | old | young | Mean |
|---|---|---|---|---|---|---|---|---|---|
| full model | **0.76**±0.01 | **0.70**±0.01 | **0.71**±0.03 | **0.71**±0.01 | **0.70**±0.02 | **0.67**±0.01 | 0.61±0.01 | **0.67**±0.01 | **0.691**±0.005 |
| $M_{\text{base}}$ | 0.75±0.02 | 0.64±0.01 | 0.68±0.01 | 0.67±0.02 | 0.66±0.01 | 0.64±0.05 | 0.60±0.02 | 0.64±0.01 | 0.660±0.007 |
| $M_{\text{no augmentation}}$ | 0.75±0.01 | 0.66±0.02 | 0.69±0.01 | 0.69±0.01 | 0.67±0.02 | 0.62±0.01 | 0.54±0.08 | 0.64±0.02 | 0.658±0.011 |
| $M_{\text{no matrix}}$ | 0.73±0.01 | 0.65±0.02 | 0.70±0.01 | 0.67±0.01 | **0.70**±0.01 | 0.65±0.01 | **0.62**±0.01 | 0.64±0.01 | 0.668±0.003 |

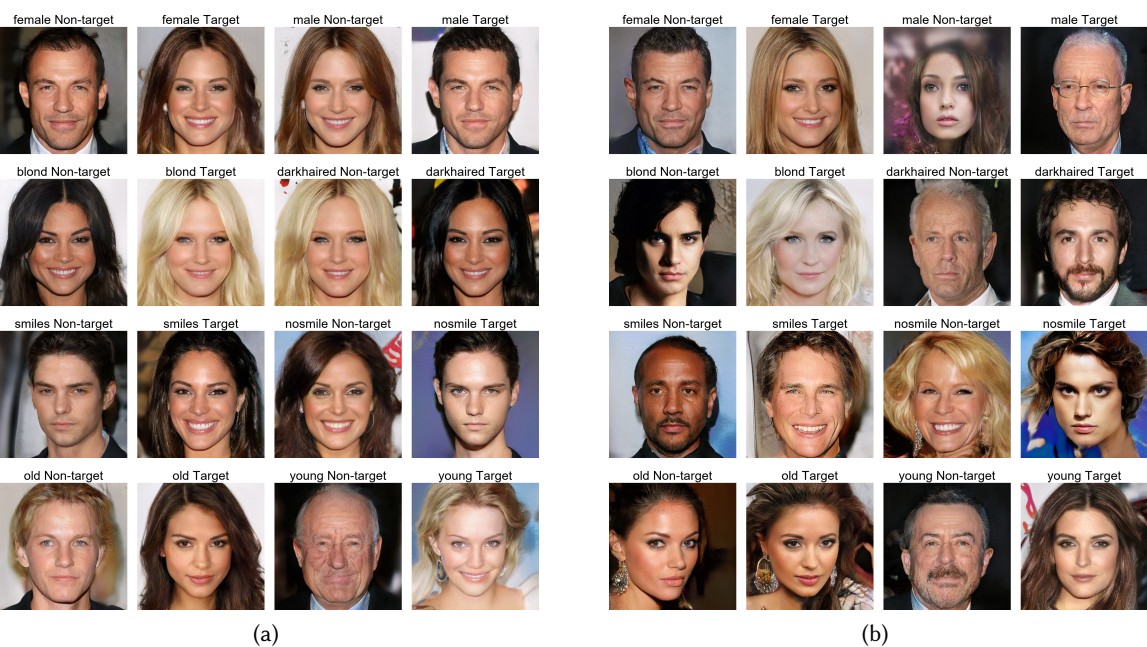

(a)                                                                                          (b)

**Figure 4: Visualization of clusters by mapping to (a) stimuli vectors from the training set and (b) randomly sampled vectors not from the training set. Each of the eight tasks results in a Target cluster and a Non-target cluster. The visualization of Target cluster should match the task description (female, male, blond, dark hair, smiles, nosmile, old, young) and Non-target visualization should not.**

Each embedding $Z$ in a cluster $C$ is mapped to one of these stimuli $v_Z = \text{argmax}_{Y \in S} \langle Z, Y \rangle$, and use the mean $M_C = \frac{1}{|C|} \sum_{Z \in C} v_Z$ to represent the cluster. We visualize it using the pre-trained generative model. For each task, we expect the image generated from $M_{C_{\text{Target}}}$ to contain salient task-specific semantic features and $M_{C_{\text{Non-target}}}$ to have the opposite semantic saliency.

**Results.** The generated images with the stimuli set and the randomly sampled set are shown in Figure 4. Between the images

from $M_{C_{\text{Target}}}$ and $M_{C_{\text{Non-target}}}$, it can be clearly seen that the semantic difference in images is correlated to the semantic task given to the participants. In Figure 4a the candidate image vectors are the stimuli vectors from the training set. In Figure 4b, 600 randomly sampled 512 vectors that are not present in the training set, are used as candidate sets and the semantic difference matching the task is still present. The salient features are clearly present across the different tasks and not present in the opposite tasks (Figure 4a).

The representations result in generated images in which the intended salient features are present even for randomly sampled candidates (Figure 4b). This indicates that the learned embeddings reflect the underlying signal from human cognition for generating task-specific salient features.

For each individual, we also visualized the subset of embeddings from a single participant similarly in supplementary materials Sec. A1.

## 4.6 Ablation Analysis

An ablation study was conducted to study the effects of participant-specific matrices and data augmentation. In contrast to the full model, three variants of the models were trained: (a) $M_{\text{no matrix}}$ that removes the participant-specific matrix; (b) $M_{\text{no augmentation}}$ that removes data augmentation; (c) $M_{\text{base}}$ that removes both.

The base model $M_{\text{base}}$ and $M_{\text{no matrix}}$ assumes that all data are collected from the same participant, and we select the cluster with the higher ERP effect as the target cluster. To minimize the differences caused by random cropping in data augmentation or other dimension changes, the base model $M_{\text{base}}$ and $M_{\text{no augmentation}}$ crops the EEG signals with fixed intervals as used in the test set.

Table 4 shows the accuracies of the full model and other variants for each task. The full model with the participant-specific matrix and augmentation consistently yields improved accuracy over all model variants in all tasks except the task old. The two variant models $M_{\text{no matrix}}$ and $M_{\text{no augmentation}}$ both have improved mean accuracy over the base model.

## 5 DISCUSSION

We presented cognitive supervision that allows to use EEG brain recordings and stimuli information to learn embeddings that capture differences in visual saliency wihtout any external labels. Below, we reflect on the two research questions we posed.

**Can representations of semantic saliency be directly learned from EEG data as a supervision signal?** We introduced a novel approach for contrastive training of models supervised solely by brain signals, demonstrating the feasibility of learning semantic visual saliencies from EEG signals. Our models successfully capture semantic saliency without relying on explicit manual annotations.

**Do the learned representations accurately capture the salient features in downstream tasks?** We evaluated the performance of our models in classification, clustering, and image generation tasks using facial image data. The results indicate enhanced performance across these tasks, comparable to classification models pre-trained and fine-tuned with extensive labeled datasets. In image generation tasks, the models demonstrate capabilities that are both valid and competitive, even when compared to models trained with manually annotated data.

**Limitations.** Currently, the field of brain-computer interfacing, while advancing, is predominantly challenged by issues of accuracy and practical convenience when compared to established user interfaces. Our experimental setup, although innovative, is principally confined to the laboratory environment and may not yet translate seamlessly to widespread public use. Nevertheless, our experiments underscore the potential to cultivate human-in-the-loop learning

systems that directly engage with human cognitive processing. These systems do not depend on manual labeling, nor do they rely on the often inaccurate and indirect manual annotations or implicit behavioral indicators. A distinct limitation of our method lies in its focus on discerning between target or non-target saliency at an individual level, essentially capturing whether an item of visual information holds salience or not. This dichotomy means our technique is not apt for annotation tasks demanding explicit labels but rather suits recognition scenarios that model individual preferences or where the task itself delineates the saliency. Examples where our method could be particularly effective include enhancing image search accuracy, CAPTCHA image detection, or in contexts where a saliency detection task is predefined for participants. Despite these limitations, numerous outcomes from our research suggest that the capabilities of models trained under our cognitive supervision framework often exceed those of models trained via conventional manual labeling. This efficacy highlights the potential of leveraging cognitive supervision to substantially refine machine learning, offering a pathway to discern human preferences toward visually salient features in a completely passive manner.

## 6 CONCLUSIONS

We have pioneered a novel demonstration that machine learning systems can be self-supervised directly from human cognitive signals captured through EEG, to detect saliency of information perceived by the user. This approach paves the way for developing machine learning systems that incorporate human-in-the-loop interactions by real-time monitoring of cognitive reactions toward digital content. While this represents a powerful new paradigm in machine learning, capable of learning user reactions to information encountered and experienced in the digital world, it also surfaces significant ethical concerns. These arise not from the recording technology per se but from the potential for broader application of cognitive signal monitoring. As technological development progresses, demonstrating that models can learn autonomously from brain responses without the need for explicit task labels or calibration, there's an increasing risk associated with the pervasive collection and use of cognitive data. Such practices, if overlooked, could enable the inference of individual and collective attitudes towards a broad spectrum of digital content, as evidenced in our initial experiments.

Therefore, the use of data must be regulated with adequate policies that can prevent lasting threats to the public. To this end, there are already actions circumventing unethical use. For instance, the EU AI act[3] prevents AI systems for the purpose of identifying or inferring emotions or intentions of natural persons on the basis of their biometric data in the workplace and educational institutions.

However, the policies are only regulating specific use cases. With this work, we call for more academic investigation to understand what is possible, theoretically and empirically, with this novel technology to formulate robust guidelines ensuring the responsible adoption and use of cognitive supervision technologies.

---

[3]https://digital-strategy.ec.europa.eu/en/policies/regulatory-framework-ai

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
