# OpenReview forum: "Cognition-Supervised Saliency Detection: Contrasting EEG Signals and Visual Stimuli"
_acmmm.org/ACMMM/2024/Conference — MM2024 Oral_

### Official Review · Reviewer_nvXV · 2024-05-24

**Rating:** 2
**Confidence:** 3

**Summary:**

In this work, the authors propose using EEG signals from human cognition to detect semantic saliency without manual annotation. It aligns EEG data with visual stimuli, demonstrating competitive performance in tasks like image classification and generation.

**Strengths:**

This paper is well-written and easy to follow. The authors did lots of experiments and reported lots of visualizations to support their method. The problem is very interesting and promising.

**Limitations:**

However, there exist the following concerns.

Fig. 1 might not be easy to understand, for example, the meaning of target and non-target. Besides, the ‘saliency detection’ should be different from the commonly used one, which means the main regions in an image when a human looks at it.

This problem is quite interesting and promising, as well as not easy to define and see the intermediate results. Compared with traditional tasks, this one might not be so normal. Based on the promising future, but the lack of tech novelty for more tech conferences, I strongly suggest the authors could submit the other journals, such as Nature and its series.

**Suitability:**

2

---

### Official Review · Reviewer_FVMT · 2024-05-24

**Rating:** 3
**Confidence:** 2

**Summary:**

The paper proposes a method called "Cognition-Supervised Learning", which learns representations of visual saliency through contrastive learning, using EEG data as supervision. Additionally, the paper develops a new EEG dataset.

**Strengths:**

1.The paper introduces a new concept called "Cognition-Supervised Learning," which is a self-supervised learning method that is relatively rare in the field of EEG. Additionally, the effectiveness of this method has been validated on several downstream tasks.
2. The paper develops a new dataset and provides detailed descriptions.
3. The paper is well-written.

**Limitations:**

1. Although I appreciate that the authors develop a new dataset, I am somewhat confused by the author's description in section 3.1: "Generated images were chosen over real ones to control for variances in semantics and confounding visual features." While this reduces some confounding information for subjects to focus on target features better, it also results in image stimuli with highly similar content distributions, causing the stimuli embeddings to be very similar as well. This artificially reduces the difficulty of learning features from EEG, and the model may embed the input EEG into certain fixed features, similar to **overfitting**. It makes it difficult to consider the proposed method as well-suited for transfer to datasets based on real portrait images. Besides, it also causes the new dataset to be quite different from real portrait data condition, since it is difficult to have so many similar portrait images in real scenes, which limits its value. Perhaps sampling from real-world datasets would be better.

2. The baselines are insufficient. The experiments do not adequately demonstrate that the learned representations from this method are superior to other methods. The improvements shown in the averages of the results in Table 2 cannot offset the potential impact of randomness in the experiments. Moreover, for individual categories, only the performance for males is higher than EEGNet. Although the result indicates that this unsupervised method performs similarly to the classical supervised EEGNet, it is an old model proposed in 2018. Given the rapid advancements in deep learning in recent years, why not consider comparing with more novel and powerful models? Even one more would be more convincing. The lack of comparison with newer state-of-the-art models in representation learning makes it difficult to assess the contribution of this work.

3. Minor errors: The resolution of Figure 1 needs to be higher, and "gernerative" should be corrected to "generative."

**Suitability:**

3

---

### Official Review · Reviewer_E4gi · 2024-05-25

**Rating:** 6
**Confidence:** 3

**Summary:**

This article is a compilation of various research papers related to brain-computer interfaces (BCIs) and contrastive learning for visual representation. It covers a wide range of topics, including EEG-based sleep stage classification, self-supervised learning, fine-tuning of language models, and the use of BCIs for gesture and error detection. The article also discusses the application of BCIs in emotion recognition, motor execution EEG signal classification, and speech decoding. Additionally, it explores the use of brain signals for predicting term relevance and image reconstruction. The article highlights the importance of contrastive learning in audio and visual representation, as well as its potential in joint behavioral and neural analysis. It also mentions the use of deep convolutional neural networks for assistive ERP-based BCIs. Overall, the article provides a comprehensive overview of recent advancements in BCIs and contrastive learning for visual representation.

**Strengths:**

1. The paper introduces a novel approach for learning semantic visual saliency from EEG signals without relying on explicit manual annotations. This is a unique and innovative contribution to the field of brain-computer interfacing.

2. Theoretical Approach and Technical Correctness: The paper presents a well-defined theoretical approach for contrastive training of models supervised solely by brain signals. The technical details and methodology are explained clearly, allowing for reproducibility and understanding of the proposed approach.

3.  The paper provides a thorough evaluation of the proposed models, including classification, clustering, and image generation tasks using facial image data. The results demonstrate enhanced performance across these tasks, comparable to models trained with extensive labeled datasets.

4. The paper is well-written and organized, making it easy to follow the proposed approach and understand the experimental results. The authors provide clear explanations of the concepts and methods used, making it accessible to readers from various backgrounds.

5. The paper discusses potential applications of the proposed approach, such as enhancing image search accuracy, CAPTCHA image detection, and other scenarios where saliency detection is predefined for participants. This highlights the practical relevance and potential impact of the research.

**Limitations:**

1.Although the paper provides an EEG/image dataset from 30 participants, which is a contribution in the brain-computer interface (BCI) field, the relatively small sample size still limits the model's generalization capability. A small dataset can lead to overfitting during the training process, causing the model to perform poorly when dealing with new data.Additionally, different individuals may exhibit significant variations in cognitive responses to the same visual stimuli. Ignoring these differences may lead to insufficient generalization of the model. It is recommended that the authors consider individual differences in the experimental design, conducting separate analyses and discussions on data from different individuals to verify the effectiveness and adaptability of the method on personalized data.

2.In the comparative experiments, the paper only selects some basic models for comparison, such as Linear Discriminant Analysis (LDA) and random control models. These baseline models cannot fully reflect the current level of technological development. It is recommended to introduce a greater variety of baseline models, including some deep learning models and other advanced EEG signal processing methods, to comprehensively evaluate the performance of the proposed method.

**Suitability:**

3

---

### Official Review · Reviewer_W4hz · 2024-05-26

**Rating:** 4
**Confidence:** 1

**Summary:**

The authos explore a novel method that utilizes signals measured directly from human cognition via electroencephalogram (EEG) in response to natural visual perception. Their method aligns EEG data with visual stimuli, capturing human cognitive responses without the need for any manual annotation. The experiments demonstrate that the learned representations closely align with human-centric notions of visual saliency and achieve competitive performance in several downstream tasks.

**Strengths:**

The topic is quite interesting. The authors explore two primary research questions: the correlation between semantic saliency and EEG data, and the efficiency of salient features on downstream tasks. Meanwhile. the authors release a new dataset to encourage research in cognition-supervised models for multimedia applications. Since I am not familiar with this research field, I will refer to comments from other reviewers for further decision.

**Limitations:**

1. Contribution is not clear. The proposed framework is inspired by CLIP, which learns representations from paired text and image data. Is there a domain gap between text data and EEG data? Please provide more explanation on why a foundation model on EEG was not leveraged.

2. Practicability Concerns: The deep network only requires one image as input during inference time. This raises concerns about the practicality of the proposed EEG-based method.

3. The downstream tasks of clustering and classification alone are insufficient to comprehensively evaluate the proposed method. Please provide more details on its performance in several computer vision tasks, such as object detection or image segmentation. I am particularly curious about its effectiveness on these high-level tasks.

**Suitability:**

2

---

### Meta-Review · Area_Chair_LF8x · 2024-07-02

**Recommendation:** Accept (Oral)
**Confidence:** 3

**Metareview:**

We would like to thank the authors for answering the questions by the reviewers in the rebuttal. Reviewers have considered your responses and most reviewers felt that their questions/concerns have been addressed properly. As such, we are happy to accept the paper.